# Fractional Multi-Step Differential Transformed Method for Approximating a Fractional Stochastic SIS Epidemic Model with Imperfect Vaccination

**DOI:** 10.3390/ijerph16060973

**Published:** 2019-03-18

**Authors:** Salah Abuasad, Ahmet Yildirim, Ishak Hashim, Samsul Ariffin Abdul Karim, J.F. Gómez-Aguilar

**Affiliations:** 1Preparatory Year Deanship, King Faisal University, 31982 Hofuf, Al-Hasa, Saudi Arabia; 2Department of Mathematics, Ege University, 35100 Bornova, Izmir, Turkey; yahmet49ege@gmail.com; 3School of Mathematical Sciences, Universiti Kebangsaan Malaysia, 43600 UKM Bangi Selangor, Malaysia; ishak_h@ukm.edu.my; 4Fundamental and Applied Sciences Department and Center for Smart Grid Energy Research (CSMER). Institute of Autonomous System, Universiti Teknologi PETRONAS, Bandar Seri Iskandar, 32610 Seri Iskandar, Perak DR, Malaysia; samsul_ariffin@utp.edu.my; 5CONACyT-Tecnológico Nacional de México/CENIDET. Interior Internado Palmira S/N, Col. Palmira, C.P. 62490 Cuernavaca, Morelos, Mexico; jgomez@cenidet.edu.mx

**Keywords:** fractional calculus, multi-step differential transformed method, differential transformed method, stochastic SIS epidemic model, imperfect vaccination

## Abstract

In this paper, we applied a fractional multi-step differential transformed method, which is a generalization of the multi-step differential transformed method, to find approximate solutions to one of the most important epidemiology and mathematical ecology, fractional stochastic SIS epidemic model with imperfect vaccination, subject to appropriate initial conditions. The fractional derivatives are described in the Caputo sense. Numerical results coupled with graphical representations indicate that the proposed method is robust and precise which can give new interpretations for various types of dynamical systems.

## 1. Introduction

Mathematical modeling of nonlinear systems is a key challenge for contemporary scientists; it is a basic description of physical reality expressed in mathematical terms. On the other hand, the study of the exact or approximate solution benefits us to understand the means of these mathematical models. Finding analytical solution is very difficult in most cases, so a good numerical solution of the problems can be gained. Fractional order derivatives provide researchers new fields for modeling numerous types of phenomena in sciences. Numerous kinds of fractional derivatives and their properties were considered (see [1,2,3,4,5,6]). The main reason for using fractional derivatives instead of integer derivatives is that fractional order model considered the memory effect, while integer order does not give us any information about the effect of the memory of human population which influences disease transmission [7].

A survey of several diverse applications which have arisen from fractional calculus is given in [4]. The most vital criteria which defined fractional derivatives was shown by Ross in [8]. Researchers are trying to find different techniques to solve linear and nonlinear fractional differential equations. Some of them modify the classical methods to be more effective, others related between two or more methods to find numerical or analytical solutions of fractional equations. For instance, in [9,10] they modified the definition of beta fractional derivative to find exact and approximate solutions of time-fractional diffusion equations in different dimensions.

Differential transform method (DTM) was firstly applied to electrical circuit problems by Zhou [11]. The DTM is well addressed in [12,13,14,15,16]. To overcome the long computations of DTM, Keskin and Oturance [17] presented an effective technique called reduced differential transform method (RDTM). Srivastava et al. [18] used RDTM for solving the (1+n)-dimensional Burgers’ equation, Yu et al. [19] applied RDTM for solving the (n+1)-dimensional case and Acan et al. [20] applied a local fractional reduced differential transform method to obtain the solutions of some linear and nonlinear partial differential equations on Cantor set. Arikoglu and Ozkol [21], Odibat et al. [22] and Odibat et al. [23] extended the algorithm of DTM to obtain analytical approximate solutions to linear and nonlinear ordinary differential equations of fractional orders.

The fractional reduced differential transform method (FRDTM) has been used successfully for solving linear and nonlinear fractional differential equations to obtain exact and approximate solutions. FRDTM was firstly introduced by Keskin and Oturance [24]. Srivastava et al. [25] studied the generalized time fractional-order biological population model (GTFBPM) by FRDTM. Rawashdeh [26] employed FRDTM to solve the nonlinear fractional Harry-Dym equation. Saravanan and Magesh [27] compared two analytical methods: FRDTM vs fractional variational iteration method (FVIM) to find numerical solutions of the linear and nonlinear Fokker-Planck partial differential equations with space and time fractional derivatives, Singh [28] presented FRDTM to compute an alternative approximate solution of initial valued autonomous system of linear and nonlinear fractional partial differential equations and Abuasad et al. [29] proposed FRDTM for finding exact and approximate solutions of the fractional Helmholtz equation.

The multi-step differential transform method (MsDTM) is presented to overcome the main drawbacks of the DTM and RDTM, which are the obtained series solution frequently converges in a very small space and the range of convergent is a very gradual process or totally divergent given a wider space. The fractional multi-step differential transform method (FMsDTM) is capable of generating approximate solutions of a wide class of linear and nonlinear problems with fractional derivatives that converge quickly to the exact solutions.

Momani et al. [30] employed FMsDTM for finding numerical approximate solutions of time-space fractional Fokker-Planck equation, Ebenezer et al. [31] applied MsDTM to fractional differential algebraic equations (FDAEs), also Ebenezer et al. [32] proposed multistep reduced differential transformation method (MRDTM) to solve linear second order telegraph equation together with time fractional derivatives, Moaddy et al. [33] presented a multistep generalized differential transform (MsGDT) to obtain accurate approximate form solution for fractional Rabinovich-Fabrikant model. Alsmadi et al. [34] proposed MRDTM for solving one-dimensional fractional heat equations with time fractional derivatives, and Arshad et al. [35] pursued the general form of FMsDTM to (n+1)-dimensional case.

The aim of this paper is to apply the FMsDTM to find an accurate approximation for a fractional stochastic SIS epidemic model with imperfect vaccination. The fractional derivatives are described in the Caputo sense. The manuscript is organized as follows: Section 2 explains the basic concepts of the Fractional Calculus; Section 3 presents the stochastic SIS epidemic model. Section 4 presents the FMsDTM. Computational illustrations are given in Section 5, Section 6 contains some basic discussions, and conclusions are given in Section 7.

## 2. Fractional Derivative

There are several definitions of fractional derivatives. The three most frequently used definitions for the general fractional fractional derivatives are: the Grünwald-Letnikov (GL) definition, the Riemann-Liouville (RL) and the Caputo definition [2,3,4]. In this paper we will use the Caputo fractional derivative, since the initial conditions for fractional order differential equations involving only the limit values of integer-order derivatives at the lower terminal initial time (t=a), such as $y^{\prime }\left(a\right),y^{\prime \prime }\left(a\right),\ldots$ [4], and also, the fractional derivative of a constant function is zero.

The Caputo fractional derivative is defined as ([4])
(1)$${}^{C}D_{a}^{\alpha }f\left(t\right):=\left\{\begin{array}{cc} \frac{1}{\Gamma (n-\alpha )}\int_{a}^{t}\frac{f^{\left(n\right)}\left(\tau \right)}{{\left(t-\tau \right)}^{\alpha +1-n}}d\tau , & n-1<\alpha <n,\\ \frac{d^{n}}{dt^{n}}f\left(t\right), & \alpha =n, \end{array}\right.$$
where α>0, t>a, n∈N and α,a,t∈R.

## 3. Stochastic SIS Epidemic Model with Imperfect Vaccination

It is essential to recognize the dynamical performance of undesirable outcomes of infectious diseases on the population development and to expect what may happen. Mathematical modeling has become a central tool in examining a varied range of such diseases to obtain a good understanding of extent mechanisms. Monitoring infectious diseases have been a progressively difficult matter and vaccination has been a normally used technique for removing illnesses for example measles, polio, diphtheria, tetanus, and tuberculosis. In addition, routine vaccination is presently provided in all developing countries against all these illnesses. It is well known that the SIS epidemic model is one of the most vital models in epidemiology and mathematical ecology. It has been thought that the immune system will create antibody against disease since vaccination doses are taken during this process. However, it may be not in a completely defensive level, specifically, the value of the vaccine is less than one. Motivated by these details, Safan and Rihan [36] considered the following SIS epidemic model with imperfect vaccination:(2)$$\left\{\begin{array}{cc} \frac{dS}{dt} & =(1-p)\mu +\alpha I-(\mu +\psi )S-\beta SI,\\ \frac{dI}{dt} & =\beta SI+(1-e)\beta VI-(\mu +\alpha )I,\\ \frac{dV}{dt} & =p\mu +\psi S-\mu V-(1-e)\beta VI, \end{array}\right.$$
subject to the initial conditions S(0), I(0) and V(0) where *S* denotes the fractions of susceptible individuals, *I* denotes the fractions of infected individuals, *V* denotes the density of vaccines who have begun the vaccination process. Individuals are assumed to be born susceptible with rate μ where a proportion *p* of them obtains vaccinated immediately after birth. Susceptible individuals can either die with rate μ, vaccinated with rate ψ or obtain infected with force of infection βI where β is the successful contact rate between infected and susceptible individuals. Infected individuals can either die with rate μ or be removed with rate α. Vaccinated individuals can either die with rate μ or obtain infected with force of infection (1−e)βI where *e* measures the efficacy of the vaccine-induced protection against infection. If e=1, then the vaccine is perfectly effective in preventing infection, while e=0 means that the vaccine has no effect. All parameter values in system (Equation 2) are assumed to be non-negative and μ>0 [37]. In system (Equation 2), the basic reproduction number $R_{v}=\frac{\beta [(1-p)\mu +(1-e)(p\mu +\psi )]}{\left(\mu +\psi )(\alpha +\mu \right)}$ is the threshold which determines whether the epidemic occurs or not. In this paper, we will discuss the case when $R_{v}\leq 1$, then system (Equation 2) has only the infection free equilibrium ${\bar{E}}_{0}=\left(\bar{S},\bar{I},\bar{V}\right)=\left(\frac{(1-p)\mu }{\mu +\psi },0,\frac{p\mu +\psi }{\mu +\psi }\right)$ and it is globally asymptotically stable in the invariant Ω, where Ω={(S,I,V):S≥0,I≥0,V≥0,S+I+V≤1}. This means that the disease will disappear and the entire population will become susceptible [37].

### Fractional Stochastic SIS Epidemic Model with Imperfect Vaccination

In this work, we modified the classical SIS epidemic model to the fractional stochastic SIS epidemic model with imperfect vaccination in the form:(3)$$\left\{\begin{array}{cc} D_{t_{0}}^{\alpha_{1}}S & =(1-p)\mu +\alpha I-(\mu +\psi )S-\beta SI,\\ D_{t_{0}}^{\alpha_{2}}I & =\beta SI+(1-e)\beta VI-(\mu +\alpha )I,\\ D_{t_{0}}^{\alpha_{3}}V & =p\mu +\psi S-\mu V-(1-e)\beta VI, \end{array}\right.$$
with the initial conditions
(4)$$S\left(t_{0}\right)=c_{1},\;I\left(t_{0}\right)=c_{2},\;V\left(t_{0}\right)=c_{3},$$
where $0<\alpha_{i}<1,i=1,2,3$, and $D_{t_{0}}^{\alpha_{i}}$ denote the Caputo fractional derivative of order $\alpha_{i}$. Indeed, if $\alpha_{i}=1,\;i=1,2,3$, then system (Equation 3) reduces to the classical system (Equation 2).

## 4. Description of the Method

The FMsDTM is a semi-numerical and analytical method and it is a modification of traditional DTM. In Table 1 we introduce the essential differential transformations of some important functions.

To illustrate the FMsDTM, consider the following system of fractional differential equations: (5)$$D_{t_{0}}^{\alpha_{i}}x_{i}\left(t\right)=h_{i}\left(t,x_{1},x_{2},\cdots ,x_{n}\right),i=1,2,\ldots ,n,\;t_{0}\leq t\leq T,$$
with the initial conditions
(6)$$x_{i}\left(t_{0}\right)=c_{i},\;i=1,2,\ldots ,n,$$
where $0<\alpha_{i}\leq 1$, $c_{i}\left(i=1,2,\ldots ,n\right)$ are real finite constants, and $D^{\alpha_{i}}$ is the Caputo fractional derivative of order $\alpha_{i}$.

Before applying the multistep method, we define the fractional differential transform of h(t) as
(7)$$H\left(k\right)=\frac{1}{\Gamma (k\alpha +1)}{\left[D_{t}^{k\alpha }h\left(t\right)\right]}_{t=t_{0}},$$
then the *n*th approximate series form solution of fractional initial value problem (FIVP) (5) and (Equation 6) can be given by
(8)$$x_{i}=\sum_{k=0}^{N}U_{i}\left(k\right){\left(t-t_{0}\right)}^{k\alpha_{i}},t\in \left[t_{0},T\right],$$
where $U_{i}$ satisfies the following recurrence relation
(9)$$U_{i}\left(K+1\right)=\frac{\Gamma (k\alpha +1)}{\Gamma (\left(k+1\right)\alpha_{i}+1)}H_{i}\left(k,U_{1},U_{2},\cdots ,U_{n}\right),\;i=1,2,\cdots ,n,$$
where $H_{i}\left(k,U_{1},U_{2},\cdots ,U_{n}\right)$ denotes the differential transformed function of $h_{i}\left(t,x_{1},x_{2},\cdots ,x_{n}\right)$, subject to the initial conditions $U_{i}\left(t_{0}\right)=c_{i},\;i=1,2,\ldots ,n$. Assume that the interval [0,T] is divided into *M* sub-intervals $[t_{i-1},t_{i}],i=1,2,\ldots ,M,$ of the same step size h=T/M, by employing the nodes $t_{i}=ih$.

The main idea of the fractional multistep DTM is explained as follows [30,31,32,33]: First of all, we apply the FDTM over the first sub-interval $\left[t_{0},t_{1}\right]$, then we can find the approximate solution, starting with $t_{0}=0,$
(10)$$x_{i,1}\left(t\right)=\sum_{k=0}^{K}U_{1}\left(k\right)t^{k\alpha_{i}},\;t\in \left[0,t_{1}\right],$$
subject to the initial conditions $x_{i,1}\left(0\right)=c_{i},\;i=1,2,\ldots ,n$. After that, for i≥2, and for each sub-interval $\left[t_{i-1},t_{i}\right]$, we will use the initial conditions $x_{i,j}\left(t_{j-1}\right)=x_{i,j-1}\left(t_{j-1}\right)$. Then, applying the FDTM to the system (5), where $t_{0}$ in (Equation 7) is replaced by $t_{i-1}$. In the same manner, we repeat the process to generate a sequence of approximate solutions $x_{i,j}\left(t\right)$, where j=1,2,⋯M, i=1,2,…,n. Therefore, the FMsDTM makes assumptions for the following solution
(11)$$x_{i}\left(t\right)=\sum_{j=1}^{M}\delta_{\nu }\;x_{i,j}\left(t\right),\;i=1,2,\ldots ,n,$$
where
$$\delta_{\nu }=\left\{\begin{array}{cc} 1, & t\in [t_{j-1},t_{j}],\\ 0, & t\notin [t_{j-1},t_{j}]. \end{array}\right.$$

The new algorithm, FMsDTM, is simple for computational performance for all values of *h*. It is easily observed that if the step size h=T, then the FMsDTM reduces to the classical FDTM. As we will see in the next section, the main advantage of the new algorithm is that the obtained solution converges for wide time regions.

## 5. Computational Illustrations

To expound the simplicity and effectiveness of the suggested method as an approximate tool for solving nonlinear problems of fractional differential equations, we apply the FMsDTM to find the approximate numerical solutions for Stochastic SIS epidemic model of fractional order (Equation 3) subject to (Equation 4). We have two main steps for MsFDTM.

### 5.1. First Step: Finding FDTM for the First Interval

We have to apply the FDTM to the system (Equation 3) by using the appropriate transformations from Table 1. We find the following recurrence relations for k≥1 over the first interval $\left[t_{0},t_{1}\right]$, where $t_{0}=0$ and $t_{1}=h$,
(12)$$\begin{array}{ccc} s_{1}\left(k\right) & = & \frac{\Gamma (\left(k-1\right)\alpha_{1}+1)}{\Gamma (k\alpha_{1}+1)}(-\beta \sum_{j=0}^{k-1}\;\left(s_{1}\left(j\right)\;i_{1}\left(k-j-1\right)\right)\\ & + & \alpha \;i_{1}\left(k-1\right)-\left(\mu +\psi \right)\;s_{1}\left(k-1\right)+\mu \left(1-p\right)),s_{1}\left(0\right)=c_{1}, \end{array}$$
(13)$$\begin{array}{ccc} i_{1}\left(k\right) & = & \frac{\Gamma (\left(k-1\right)\alpha_{2}+1)}{\Gamma (k\alpha_{2}+1)}(\beta \left(1-e\right)\sum_{\xi =0}^{k-1}\left(v_{1}\left(\xi \right)\;i_{1}\left(k-\xi -1\right)\right)\\ & + & \beta \sum_{j=0}^{k-1}(s_{1}\left(j\right)\;i_{1}\left(k-j-1\right))-\left(\alpha +\mu \right)\;i_{1}\left(k-1\right)),i_{1}\left(0\right)=c_{2}, \end{array}$$
(14)$$\begin{array}{ccc} v_{1}\left(k\right) & = & \frac{\Gamma (\left(k-1\right)\alpha_{3}+1)}{\Gamma (k\alpha_{3}+1)}(-\beta \left(1-e\right)\sum_{\theta =0}^{k-1}\left(v_{1}\left(\theta \right)\;i_{1}\left(k-\theta -1\right)\right)\\ & + & \psi \;s_{1}\left(k-1\right)-\mu \;v_{1}\left(k-1\right)+\mu p),v_{1}\left(0\right)=c_{3}, \end{array}$$
where $s_{1}\left(k\right)$, $i_{1}\left(k\right)$ and $v_{1}\left(k\right)$ are the transformed functions of the $S_{1}\left(t\right)$, $I_{1}\left(t\right)$ and $V_{1}\left(t\right)$, respectively. With FDTM of initial conditions of the form $s_{1}\left(0\right)=c_{1}$, $i_{1}\left(0\right)=c_{2}$ and $v_{1}\left(0\right)=c_{3}$. The process generates a sequence of approximate solutions such that
(15)$$S_{1}\left(t\right):=\sum_{n=0}^{K}s_{1}\left(n\right)t^{\alpha_{1}n},$$
(16)$$I_{1}\left(t\right):=\sum_{n=0}^{K}i_{1}\left(n\right)t^{\alpha_{2}n},$$
(17)$$V_{1}\left(t\right):=\sum_{n=0}^{K}v_{1}\left(n\right)t^{\alpha_{3}n}.$$


### 5.2. Second Step: Apply the MsFDTM for All Intervals

We have to repeat the first step (Section 5.1) to all sub-intervals $\left[t_{r-1},t_{r}\right]$ where $t_{r}=hr$, for all r≥2, using the initial conditions $s_{r}\left(0\right)=S_{r}\left(t_{r-1}\right)=S_{r-1}\left(t_{r-1}\right)$, $i_{r}\left(0\right)=I_{r}\left(t_{r-1}\right)=I_{r-1}\left(t_{r-1}\right)$ and $v_{r}\left(0\right)=V_{r}\left(t_{r-1}\right)=V_{r-1}\left(t_{r-1}\right)$. The process generates a sequence of approximate solutions such that
(18)$$S_{r}\left(t\right):=\sum_{n=0}^{K}s_{r}\left(n\right){\left(t-t_{r-1}\right)}^{\alpha_{1}n},$$
(19)$$I_{r}\left(t\right):=\sum_{n=0}^{K}i_{r}\left(n\right){\left(t-t_{r-1}\right)}^{\alpha_{2}n},$$
(20)$$V_{r}\left(t\right):=\sum_{n=0}^{K}v_{r}\left(n\right){\left(t-t_{r-1}\right)}^{\alpha_{3}n},$$
therefore, the multistep approximate series solutions of the system (Equation 3) can be given as
$$S\left(t\right)=\left\{\begin{array}{cc} S_{1}\left(t\right)=\sum_{n=0}^{K}s_{1}\left(n\right)t^{n\alpha_{1}} & ,t\in [0,t_{1}],\\ S_{2}\left(t\right)=\sum_{n=0}^{K}s_{2}\left(n\right){\left(t-t_{1}\right)}^{n\alpha_{1}} & ,t\in [t_{1},t_{2}],\\ \vdots \\ S_{M}\left(t\right)=\sum_{n=0}^{K}s_{M}\left(n\right){\left(t-t_{M-1}\right)}^{n\alpha_{1}} & ,t\in [t_{M-1},t_{M}]. \end{array}\right.$$
$$I\left(t\right)=\left\{\begin{array}{cc} I_{1}\left(t\right)=\sum_{n=0}^{K}i_{1}\left(n\right)t^{n\alpha_{2}} & ,t\in [0,t_{1}],\\ I_{2}\left(t\right)=\sum_{n=0}^{K}i_{2}\left(n\right){\left(t-t_{1}\right)}^{n\alpha_{2}} & ,t\in [t_{1},t_{2}],\\ \vdots \\ I_{M}\left(t\right)=\sum_{n=0}^{K}i_{M}\left(n\right){\left(t-t_{M-1}\right)}^{n\alpha_{2}} & ,t\in [t_{M-1},t_{M}]. \end{array}\right.$$
$$V\left(t\right)=\left\{\begin{array}{cc} V_{1}\left(t\right)=\sum_{n=0}^{K}v_{1}\left(n\right)t^{n\alpha_{3}} & ,t\in [0,t_{1}],\\ V_{2}\left(t\right)=\sum_{n=0}^{K}v_{2}\left(n\right){\left(t-t_{1}\right)}^{n\alpha_{3}} & ,t\in [t_{1},t_{2}],\\ \vdots \\ V_{M}\left(t\right)=\sum_{n=0}^{K}v_{M}\left(n\right){\left(t-t_{M-1}\right)}^{n\alpha_{3}} & ,t\in [t_{M-1},t_{M}]. \end{array}\right.$$
where $t\in [t_{r-1},t_{r}]$, $v_{r}\left(k\right)$, $i_{r}\left(k\right)$ and $v_{r}\left(k\right)$ for r=1,2,⋯,M satisfy the following recurrence relations
(21)$$\begin{array}{ccc} s_{r}\left(k\right) & = & \frac{\Gamma (\left(k-1\right)\alpha_{1}+1)}{\Gamma (k\alpha_{1}+1)}(-\beta \sum_{j=0}^{k-1}\;\left(s_{r}\left(j\right)\;i_{r}\left(-j+k-1\right)\right)\\ & + & \alpha \;i_{r}\left(k-1\right)-\left(\mu +\psi \right)\;s_{r}\left(k-1\right)+\mu \left(1-p\right)), \end{array}$$
(22)$$\begin{array}{ccc} i_{r}\left(k\right) & = & \frac{\Gamma (\left(k-1\right)\alpha_{2}+1)}{\Gamma (k\alpha_{2}+1)}(\beta \left(1-e\right)\sum_{\xi =0}^{k-1}\left(v_{r}\left(\xi \right)\;i_{r}\left(k-\xi -1\right)\right)\\ & + & \beta \sum_{j=0}^{k-1}(s_{r}\left(j\right)\;i_{r}\left(-j+k-1\right))-\left(\alpha +\mu \right)\;i_{r}\left(k-1\right)), \end{array}$$
(23)$$\begin{array}{ccc} v_{r}\left(k\right) & = & \frac{\Gamma (\left(k-1\right)\alpha_{3}+1)}{\Gamma (k\alpha_{3}+1)}(-\beta \left(1-e\right)\sum_{\theta =0}^{k-1}\left(v_{r}\left(\theta \right)\;i_{r}\left(-\theta +k-1\right)\right)\\ & + & \psi \;s_{r}\left(k-1\right)-\mu \;v_{r}\left(k-1\right)+\mu p), \end{array}$$
where $s_{r}\left(k\right)$, $i_{r}\left(k\right)$ and $v_{r}\left(k\right)$ are the transformed functions of the S(t), I(t) and V(t), respectively. The most confusing issue in this method is finding the initial conditions. Starting with the given initial condition say $s_{1}\left(0\right)=c_{1}$, then $s_{2}\left(0\right)=S_{2}\left(t_{1}\right)=S_{1}\left(t_{1}\right)$ that is we have to substitute $t_{1}=h$ in $S_{1}\left(t\right)$, then to find $s_{3}\left(0\right)$ we have to substitute $t_{2}=2h$ in $S_{2}\left(t\right)$ and so on, for the last term, to find $s_{100}\left(0\right)$ we have to substitute $t_{99}=99h$ in $S_{99}\left(t\right)$. We follow same steps for finding the initial conditions for $i_{r}\left(0\right)$ and $v_{r}\left(0\right)$, r=1,2,…,M−1.

To apply the multistep fractional differential equations we take three cases for different values of $\alpha_{i},i=\left\{1,2,3\right\}$, where T=20,M=100,K=10, h=T/M and N=1000. We use parameter values from Table 2 for the numerical calculations.

### 5.3. Case 1: $\alpha_{i}=1,\;i=\left\{1,2,3\right\}$. (Non-Fractional Case)

Firstly, we apply step (Section 5.1) by finding the recurrence relation (Equation 12), (22) and (23) for 1≤k≤K over the first interval $\left[0,t_{1}\right]$, where $t_{1}=h=0.2$. If we start with the initial conditions $s_{1}\left(0\right)=0.1$, $i_{1}\left(0\right)=0.05$ and $v_{1}\left(0\right)=0.5$, then we can find a sequence of approximate solutions (Equation 15), (Equation 16) and (Equation 17).

Secondly, we have to apply step (Section 5.2) for all intervals $[t_{i-1},t_{i}],$ where 2≤i≤M and t(i)=hi. Therefore, the multistep approximate series solutions of the system (Equation 3) can be given by
$$S\left(t\right)=\left\{\begin{array}{cc} S_{1}\left(t\right)= & 0.000984869t^{10}+0.00109228t^{9}+0.00122592t^{8}+0.00139671t^{7}\\ & +0.00162254t^{6}+0.00193517t^{5}+0.002392t^{4}+0.00318393t^{3}\\ & +0.00408125t^{2}+0.0055t+0.1,\;t\in \left[0,0.2\right],\\ S_{2}\left(t\right)= & 0.00098496{\left(t-0.2\right)}^{10}+0.00109239{\left(t-0.2\right)}^{9}+0.00122606{\left(t-0.2\right)}^{8}\\ & +0.00139689{\left(t-0.2\right)}^{7}+0.00162279{\left(t-0.2\right)}^{6}+0.00193548{\left(t-0.2\right)}^{5}\\ & +0.00239293{\left(t-0.2\right)}^{4}+0.00317968{\left(t-0.2\right)}^{3}+0.00413623{\left(t-0.2\right)}^{2}\\ & +0.0051105(t-0.2)+0.101293,\;t\in [0.2,0.4],\\ \vdots & \\ S_{100}\left(t\right)= & 0.00098764{\left(t-19.8\right)}^{10}+0.00109569{\left(t-19.8\right)}^{9}+0.00123024{\left(t-19.8\right)}^{8}\\ & +0.00140232{\left(t-19.8\right)}^{7}+0.00163014{\left(t-19.8\right)}^{6}+0.00194571{\left(t-19.8\right)}^{5}\\ & +0.00241183{\left(t-19.8\right)}^{4}+0.00314288{\left(t-19.8\right)}^{3}+0.00507408{\left(t-19.8\right)}^{2}\\ & -0.00179874(t-19.8)+0.112344,\;t\in [19.8,20]. \end{array}\right.$$
$$I\left(t\right)=\left\{\begin{array}{cc} I_{1}\left(t\right)= & 8.97489\times 10^{-6}t^{10}+0.0000111362t^{9}+0.0000141775t^{8}+0.0000186435t^{7}\\ & +0.0000255772t^{6}+0.0000370728t^{5}+0.0000586803t^{4}+0.0000781175t^{3}\\ & +0.0008675t^{2}-0.008t+0.05,\;t\in \left[0,0.2\right],\\ I_{2}\left(t\right)= & 8.707713\times 10^{-6}{\left(t-0.2\right)}^{10}+0.0000108052{\left(t-0.2\right)}^{9}+0.0000137567{\left(t-0.2\right)}^{8}\\ & +0.0000180914{\left(t-0.2\right)}^{7}+0.000024822{\left(t-0.2\right)}^{6}+0.0000359864{\left(t-0.2\right)}^{5}\\ & +0.0000568806{\left(t-0.2\right)}^{4}+0.0000773632{\left(t-0.2\right)}^{3}+0.000827064{\left(t-0.2\right)}^{2}\\ & -0.00768524(t-0.2)+0.0484354,\;t\in [0.2,0.4],\\ \vdots & \\ I_{100}\left(t\right)= & 2.56076782\times 10^{-6}{\left(t-19.8\right)}^{6}+3.73407769\times 10^{-6}{\left(t-19.8\right)}^{5}\\ & +5.82758646\times 10^{-6}{\left(t-19.8\right)}^{4}+0.0000125572{\left(t-19.8\right)}^{3}\\ & +0.0000306418{\left(t-19.8\right)}^{2}-0.000457295\left(t-19.8\right)+0.00460685,\;t\in \left[19.8,20\right]. \end{array}\right.$$
$$V\left(t\right)=\left\{\begin{array}{cc} V_{1}\left(t\right)= & 0.00889641t^{10}+0.00987076t^{9}+0.0110843t^{8}+0.0126369t^{7}+0.0146933t^{6}\\ & +0.0175436t^{5}+0.0217562t^{4}+0.028463t^{3}+0.0433013t^{2}+0.0375t+0.5,\;t\in \left[0,0.2\right],\\ V_{2}\left(t\right)= & 0.00889659{\left(t-0.2\right)}^{10}+0.00987098{\left(t-0.2\right)}^{9}+0.0110845{\left(t-0.2\right)}^{8}\\ & +0.0126373{\left(t-0.2\right)}^{7}+0.0146938{\left(t-0.2\right)}^{6}+0.0175444{\left(t-0.2\right)}^{5}\\ & +0.0217573{\left(t-0.2\right)}^{4}+0.0284644{\left(t-0.2\right)}^{3}+0.0433451{\left(t-0.2\right)}^{2}\\ & +0.0366079(t-0.2)+0.509501,\;t\in [0.2,0.4],\\ \vdots \\ V_{100}\left(t\right)= & 0.00890173{\left(t-19.8\right)}^{10}+0.00987737{\left(t-19.8\right)}^{9}+0.0110927{\left(t-19.8\right)}^{8}\\ & +0.0126481{\left(t-19.8\right)}^{7}+0.0147087{\left(t-19.8\right)}^{6}+0.0175664{\left(t-19.8\right)}^{5}\\ & +0.0217911{\left(t-19.8\right)}^{4}+0.0285003{\left(t-19.8\right)}^{3}+0.0451649{\left(t-19.8\right)}^{2}\\ & -0.00286168(t-19.8)+0.933828,\;t\in [19.8,20]. \end{array}\right.$$


### 5.4. Case 2: $\alpha_{i}=0.7,i=\left\{1,2,3\right\}$

We also apply step (Section 5.1) by finding the recurrence relation (Equation 12), (22) and (23) for 1≤k≤K over the first interval $\left[0,t_{1}\right]$, where $t_{1}=h=0.2$. If we start with the initial conditions $s_{1}\left(0\right)=0.06,i_{1}\left(0\right)=0.1$ and $v_{1}\left(0\right)=0.5$, then we can find a sequence of of approximate solutions (Equation 15), (Equation 16) and (Equation 17).

Then, we have to apply step (Section 5.2) for all intervals $[t_{i-1},t_{i}],$ where 2≤i≤M and t(i)=hi.

Therefore, the multistep approximate series solutions of the system (Equation 3) can be given by
$$S\left(t\right)=\left\{\begin{array}{cc} S_{1}\left(t\right)= & 0.000625242t^{4.9}+0.000690271t^{4.2}+0.00144703t^{2.1}+0.000529103t^{6.3}\\ & +0.000572176t^{5.6}+0.000787145t^{3.5}+0.000843143t^{2.8}-0.00119881t^{1.4}\\ & +0.0176088t^{0.7}+0.000493134t^{7}+0.06,\;t\in \left[0,0.2\right],\\ S_{2}\left(t\right)= & 0.000493376{\left(t-0.2\right)}^{7}+0.000529377{\left(t-0.2\right)}^{6.3}+0.000572494{\left(t-0.2\right)}^{5.6}\\ & +0.000625586{\left(t-0.2\right)}^{4.9}+0.000690843{\left(t-0.2\right)}^{4.2}+0.00078681{\left(t-0.2\right)}^{3.5}\\ & +0.0008491{\left(t-0.2\right)}^{2.8}+0.00141452{\left(t-0.2\right)}^{2.1}-0.000989761{\left(t-0.2\right)}^{1.4}\\ & +0.016367{\left(t-0.2\right)}^{0.7}+0.0656441,\;t\in \left[0.2,0.4\right],\\ \vdots & \\ S_{100}\left(t\right)= & 0.000500026{\left(t-19.8\right)}^{7}+0.000536991{\left(t-19.8\right)}^{6.3}+0.000581379{\left(t-19.8\right)}^{5.6}\\ & +0.000635877{\left(t-19.8\right)}^{4.9}+0.000704688{\left(t-19.8\right)}^{4.2}+0.000794965{\left(t-19.8\right)}^{3.5}\\ & +0.000918927{\left(t-19.8\right)}^{2.8}+0.00110623{\left(t-19.8\right)}^{2.1}+0.00147171{\left(t-19.8\right)}^{1.4}\\ & -0.00155616{\left(t-19.8\right)}^{0.7}+0.13312,\;t\in \left[19.8,20\right]. \end{array}\right.$$
$$I\left(t\right)=\left\{\begin{array}{cc} I_{1}\left(t\right)= & 0.0000377215t^{4.9}+0.0000480147t^{4.2}-0.000135681t^{2.1}+0.0000266828t^{6.3}\\ & +0.0000314804t^{5.6}+0.0000539636t^{3.5}+0.000123516t^{2.8}+0.00185868t^{1.4}\\ & -0.0118859t^{0.7}+0.000023039t^{7}+0.1,\;t\in \left[0,0.2\right],\\ I_{2}\left(t\right)= & 0.0000222095{\left(t-0.2\right)}^{7}+0.0000257227{\left(t-0.2\right)}^{6.3}+0.0000303485{\left(t-0.2\right)}^{5.6}\\ & +0.0000363664{\left(t-0.2\right)}^{4.9}+0.0000462921{\left(t-0.2\right)}^{4.2}+0.0000520346{\left(t-0.2\right)}^{3.5}\\ & +0.000119067{\left(t-0.2\right)}^{2.8}-0.000130398{\left(t-0.2\right)}^{2.1}+0.00178731{\left(t-0.2\right)}^{1.4}\\ & -0.0114139{\left(t-0.2\right)}^{0.7}+0.0963397,\;t\in \left[0.2,0.4\right],\\ \vdots & \\ I_{100}\left(t\right)= & 1.33552438\times 10^{-6}{\left(t-19.8\right)}^{7}+1.549701353\times 10^{-6}{\left(t-19.8\right)}^{6.3}\\ & +1.831903793\times 10^{-6}{\left(t-19.8\right)}^{5.6}+2.2060142014\times 10^{-6}{\left(t-19.8\right)}^{4.9}\\ & +2.79235825\times 10^{-6}{\left(t-19.8\right)}^{4.2}+3.32778156\times 10^{-6}{\left(t-19.8\right)}^{3.5}\\ & +6.42780969\times 10^{-6}{\left(t-19.8\right)}^{2.8}-2.1166521\times 10^{-6}{\left(t-19.8\right)}^{2.1}\\ & +0.0000681475{\left(t-19.8\right)}^{1.4}-0.00038077{\left(t-19.8\right)}^{0.7}+0.00542622,\;t\in \left[19.8,20\right], \end{array}\right.$$
$$V\left(t\right)=\left\{\begin{array}{cc} V_{1}\left(t\right)= & 0.00572536t^{4.9}+0.00634544t^{4.2}+0.00982918t^{2.1}+0.00483486t^{6.3}\\ & +0.00523462t^{5.6}+0.00715549t^{3.5}+0.00829221t^{2.8}+0.0138605t^{1.4}\\ & +0.00176088t^{0.7}+0.00450196t^{7}+0.5,\;t\in \left[0,0.2\right],\\ V_{2}\left(t\right)= & 0.00450253{\left(t-0.2\right)}^{7}+0.00483551{\left(t-0.2\right)}^{6.3}+0.0052354{\left(t-0.2\right)}^{5.6}\\ & +0.0057263{\left(t-0.2\right)}^{4.9}+0.0063466{\left(t-0.2\right)}^{4.2}+0.00715712{\left(t-0.2\right)}^{3.5}\\ & +0.00829366{\left(t-0.2\right)}^{2.8}+0.00983769{\left(t-0.2\right)}^{2.1}+0.0138246{\left(t-0.2\right)}^{1.4}\\ & +0.00201324{\left(t-0.2\right)}^{0.7}+0.502489,\;t\in \left[0.2,0.4\right],\\ \vdots & \\ V_{100}\left(t\right)= & 0.00451675{\left(t-19.8\right)}^{7}+0.00485207{\left(t-19.8\right)}^{6.3}+0.00525503{\left(t-19.8\right)}^{5.6}\\ & +0.00575018{\left(t-19.8\right)}^{4.9}+0.00637622{\left(t-19.8\right)}^{4.2}+0.00719794{\left(t-19.8\right)}^{3.5}\\ & +0.00833505{\left(t-19.8\right)}^{2.8}+0.0100239{\left(t-19.8\right)}^{2.1}+0.0131474{\left(t-19.8\right)}^{1.4}\\ & +0.00263195{\left(t-19.8\right)}^{0.7}+0.815991,\;t\in \left[19.8,20\right]. \end{array}\right.$$


### 5.5. Case 3: $\alpha_{i}=0.5,i=\left\{1,2,3\right\}$

Again, we apply step (Section 5.1) by finding the recurrence relation (Equation 12), (22) and (23) for 1≤k≤K over the first interval $\left[0,t_{1}\right]$, where $t_{1}=h=0.2$. If we start with the initial conditions $s_{1}\left(0\right)=0.06$, $i_{1}\left(0\right)=0.1$ and $v_{1}\left(0\right)=0.5$, then we can find a sequence of of approximate solutions (Equation 15), (Equation 16) and (Equation 17).

Then, we have to apply step (Section 5.2) for all intervals $[t_{i-1},t_{i}],$ where 2≤i≤M and t(i)=hi. Therefore, the multistep approximate series solutions of the system (Equation 3) can be given by
(24)$$S\left(t\right)=\left\{\begin{array}{cc} S_{1}\left(t\right)= & 0.000443468t^{4.5}+0.000497927t^{3.5}+0.000596398t^{2.5}+0.00123275t^{1.5}\\ & +0.0186183t^{0.5}+0.000422425t^{5}+0.000467803t^{4}+0.000528807t^{3}\\ & +0.000537843t^{2}-0.00210027t+0.05,\;t\in \left[0,0.2\right],\\ S_{2}\left(t\right)= & 0.000422758{\left(t-0.2\right)}^{5}+0.00044382{\left(t-0.2\right)}^{4.5}+0.000468206{\left(t-0.2\right)}^{4}\\ & +0.000498249{\left(t-0.2\right)}^{3.5}+0.000529695{\left(t-0.2\right)}^{3}+0.000594742{\left(t-0.2\right)}^{2.5}\\ & +0.000548902{\left(t-0.2\right)}^{2}+0.00117982{\left(t-0.2\right)}^{1.5}-0.00181658\left(t-0.2\right)\\ & +0.0170407{\left(t-0.2\right)}^{0.5}+0.0580559,\;t\in \left[0.2,0.4\right],\\ \vdots & \\ S_{100}\left(t\right)= & 0.000431831{\left(t-19.8\right)}^{5}+0.000453682{\left(t-19.8\right)}^{4.5}+0.000479227{\left(t-19.8\right)}^{4}\\ & +0.000509647{\left(t-19.8\right)}^{3.5}+0.000546677{\left(t-19.8\right)}^{3}+0.000593318{\left(t-19.8\right)}^{2.5}\\ & +0.000653399{\left(t-19.8\right)}^{2}+0.000740852{\left(t-19.8\right)}^{1.5}+0.000898019\left(t-19.8\right)\\ & -0.00151172{\left(t-19.8\right)}^{0.5}+0.135435,\;t\in \left[19.8,20\right]. \end{array}\right.$$
(25)$$I\left(t\right)=\left\{\begin{array}{cc} I_{1}\left(t\right)= & 0.0000374582t^{4.5}+0.0000470116t^{3.5}+0.0000513417t^{2.5}\\ & -0.000241316t^{1.5}-0.0102683t^{0.5}+0.0000338951t^{5}+0.0000420655t^{4}\\ & +0.0000579396t^{3}+0.00014912t^{2}+0.0019145t+0.1,\;t\in \left[0,0.2\right],\\ I_{2}\left(t\right)= & 0.0000324746{\left(t-0.2\right)}^{5}+0.0000358888{\left(t-0.2\right)}^{4.5}+0.0000403047{\left(t-0.2\right)}^{4}\\ & +0.0000450407{\left(t-0.2\right)}^{3.5}+0.0000555313{\left(t-0.2\right)}^{3}+0.0000491304{\left(t-0.2\right)}^{2.5}\\ & +0.000143186{\left(t-0.2\right)}^{2}-0.00023246{\left(t-0.2\right)}^{1.5}+0.00183863\left(t-0.2\right)\\ & -0.00982993{\left(t-0.2\right)}^{0.5}+0.0957768,\;t\in \left[0.2,0.4\right],\\ \vdots & \\ I_{100}\left(t\right)= & 1.2777073\times 10^{-6}{\left(t-19.8\right)}^{5}+1.41492377\times 10^{-6}{\left(t-19.8\right)}^{4.5}\\ & +1.5905276\times 10^{-6}{\left(t-19.8\right)}^{4}+1.7915366\times 10^{-6}{\left(t-19.8\right)}^{3.5}\\ & +2.1705987\times 10^{-6}{\left(t-19.8\right)}^{3}+2.14506543\times 10^{-6}{\left(t-19.8\right)}^{2.5}\\ & +4.8382885\times 10^{-6}{\left(t-19.8\right)}^{2}-4.7291108\times 10^{-6}{\left(t-19.8\right)}^{1.5}\\ & +0.0000496015\left(t-19.8\right)-0.000196208{\left(t-19.8\right)}^{0.5}+0.00350754,\;t\in \left[19.8,20\right]. \end{array}\right.$$
(26)$$V\left(t\right)=\left\{\begin{array}{cc} V_{1}\left(t\right)= & 0.00408213t^{4.5}+0.00458497t^{3.5}+0.00533058t^{2.5}\\ & +0.00646462t^{1.5}-0.00552906t^{0.5}+0.00388567t^{5}+0.00431182t^{4}\\ & +0.00491906t^{3}+0.00591018t^{2}+0.00902304t+0.6,\;t\in \left[0,0.2\right],\\ V_{2}\left(t\right)= & 0.00388668{\left(t-0.2\right)}^{5}+0.00408324{\left(t-0.2\right)}^{4.5}+0.00431308{\left(t-0.2\right)}^{4}\\ & +0.00458642{\left(t-0.2\right)}^{3.5}+0.00492069{\left(t-0.2\right)}^{3}+0.00533289{\left(t-0.2\right)}^{2.5}\\ & +0.00591114{\left(t-0.2\right)}^{2}+0.0064773{\left(t-0.2\right)}^{1.5}+0.00896389\left(t-0.2\right)\\ & -0.00504602{\left(t-0.2\right)}^{0.5}+0.600309,\;t\in \left[0.2,0.4\right],\\ \vdots & \\ V_{100}\left(t\right)= & 0.00390885{\left(t-19.8\right)}^{5}+0.00410791{\left(t-19.8\right)}^{4.5}+0.00434082{\left(t-19.8\right)}^{4}\\ & +0.00461838{\left(t-19.8\right)}^{3.5}+0.00495695{\left(t-19.8\right)}^{3}+0.00538236{\left(t-19.8\right)}^{2.5}\\ & +0.00594071{\left(t-19.8\right)}^{2}+0.00670944{\left(t-19.8\right)}^{1.5}+0.00797443\left(t-19.8\right)\\ & +0.00107813{\left(t-19.8\right)}^{0.5}+0.895895,\;t\in \left[19.8,20\right]. \end{array}\right.$$


## 6. Discussion

According to Section 3 and Table 2, we give the following discussions:

Figure 1 depicts Case (Section 5.3): $\alpha_{j}=1,j=\left\{1,2,3\right\}$ in Table 3 where the invariant $\Omega_{j}\leq 1$ for all j=0,1,…,14. The basic reproduction number $R_{v}=0.636364<1$ and the infection free equilibrium $\bar{E_{0}}=\left(\bar{S},\bar{I},\bar{V}\right)=\left(0.0909091,0,0.909091\right)$ is globally asymptotically stable, so, from t=0 to t=14 we can conclude for the non-fractional stochastic SIS epidemic model with imperfect vaccination (Equation 3) that the disease will disappear and the entire population will become susceptible.

Also, Figure 2 depicts Case (Section 5.4): $\alpha_{j}=0.7,j=\left\{1,2,3\right\}$ in Table 4 where the invariant $\Omega_{j}\leq 1$ for all j=0,1,…,20. The basic reproduction number $R_{v}=0.806061<1$ and the infection free equilibrium $\bar{E_{0}}=\left(\bar{S},\bar{I},\bar{V}\right)=\left(0.0666667,0,0.933333\right)$ is globally asymptotically stable, so, from t=0 to t=20 we can conclude for the fractional stochastic SIS epidemic model with imperfect vaccination (Equation 3) that the disease will disappear and the entire population will become susceptible.

Finally, Figure 3 depicts Case (Section 5.5): $\alpha_{j}=0.5,j=\left\{1,2,3\right\}$ in Table 5 where the invariant $\Omega_{j}\leq 1$ for all j=0,1,…,17. The basic reproduction number $R_{v}=0.8<1$ and the infection free equilibrium $\bar{E_{0}}=\left(\bar{S},\bar{I},\bar{V}\right)=\left(0.05,0,0.95\right)$ is globally asymptotically stable, so, from t=0 to t=17 we can conclude for the fractional stochastic SIS epidemic model with imperfect vaccination (Equation 3) that the disease will disappear and the entire population will become susceptible.

## 7. Conclusions

Building fractional mathematical models for physical phenomenon, as well as developing numerical and analytical solutions for such models are very important issue in epidemiology and mathematical ecology. In this work, the so-called FMsDTM is successfully applied in handling stochastic SIS epidemic model of fractional-order. Numerical results together with graphical representations show the total reliability and effectiveness of the proposed technique with a vast potential in scientific applications. The method works successfully in handling systems of differential equations directly with a minimum size of computations and a wide interval of convergence for the series solution. Also, the method reduces the computational difficulties of the other methods and all the calculations can be made by simple manipulations.

The approximate solutions using FMsDTM of fractional Stochastic SIS epidemic model suggest new and promising interpretations for ecological systems more than the integer-order systems, and this is actually one of the main reasons for generalizing the integer-order differential equations to fractional-order differential equations.

## Figures and Tables

**Figure 1 ijerph-16-00973-f001:**
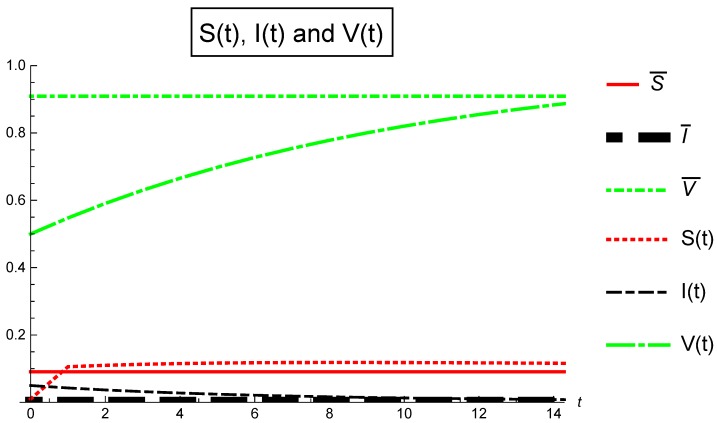
Population fraction verses time for Case 1: $\alpha_{1}=\alpha_{2}=\alpha_{3}=1$.

**Figure 2 ijerph-16-00973-f002:**
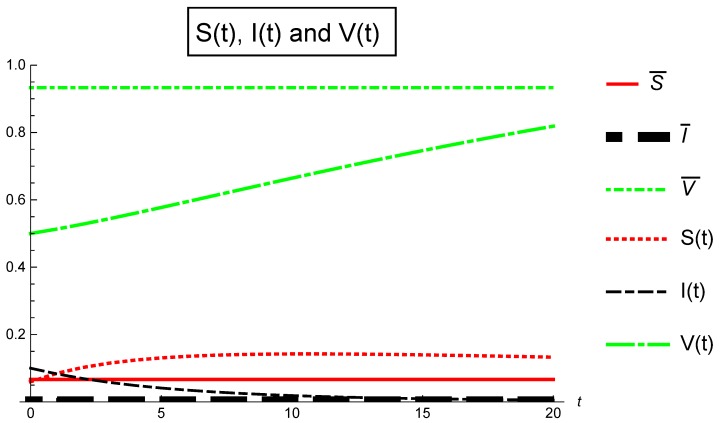
Population fraction verses time for Case 2: $\alpha_{1}=\alpha_{2}=\alpha_{3}=0.7$.

**Figure 3 ijerph-16-00973-f003:**
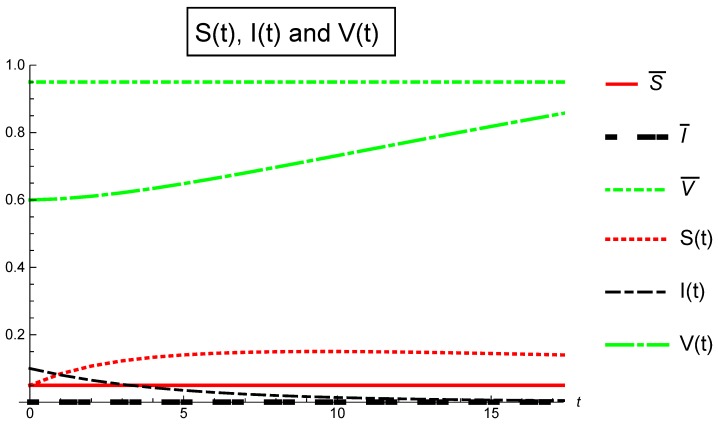
Population fraction verses time for Case 3: $\alpha_{1}=\alpha_{2}=\alpha_{3}=0.5$.

**Table 1 ijerph-16-00973-t001:** Differential Transformations [15,17,29,35].

Original Function	Transformed Function
h(t)=f(t)±g(t)	H(k)=F(k)±G(k)
h(t)=λf(t)	H(k)=λF(k)
h(t)=f(t)g(t)	$H\left(k\right)=\sum_{n=0}^{k}F\left(n\right)G\left(k-n\right)$
$h\left(t\right)=D_{t}^{n\alpha }f\left(t\right)$	$H\left(k\right)=\frac{\Gamma (k\alpha +n\alpha +1)}{\Gamma (k\alpha +1)}F\left(k+n\right)$
$h\left(t\right)=t^{n}$	H(k)=δ(k−n)
h(t)=exp(λt)	$H\left(k\right)=\lambda^{k}/k!$
h(t)=sin(ωt+α)	$H\left(k\right)=\left(\omega^{k}/k!\right)\sin\left(\pi k/2+\alpha \right)$
h(t)=cos(ωt+α)	$H\left(k\right)=\left(\omega^{k}/k!\right)\cos\left(\pi k/2+\alpha \right)$

**Table 2 ijerph-16-00973-t002:** Different parameter values used for SIS epidemic model.

Case	$\alpha_{i}$	*p*	μ	β	α	*e*	ψ	$R_{v}$	Comments
1	1	0.9	0.1	0.7	0.2	0.8	0.01	0.636364	${\bar{E}}_{0}$ stable
2	0.7	0.9	0.02	0.7	0.2	0.8	0.01	0.806061	${\bar{E}}_{0}$ stable
3	0.5	0.9	0.01	0.7	0.2	0.8	0.01	0.8	${\bar{E}}_{0}$ stable

**Table 3 ijerph-16-00973-t003:** $\alpha_{1}=\alpha_{2}=\alpha_{3}=1$.

*t*	S(t)	I(t)	V(t)	Ω
0.	0.05	0.1	0.6	0.75
1.	0.105749	0.0427621	0.545775	0.694286
2.	0.109974	0.0367964	0.587467	0.734237
3.	0.113037	0.03184	0.625454	0.770331
4.	0.115206	0.0276914	0.660073	0.802971
5.	0.116686	0.0241951	0.691627	0.832508
6.	0.117631	0.0212297	0.720388	0.859249
7.	0.118162	0.0186994	0.746603	0.883465
8.	0.118374	0.0165286	0.770498	0.9054
9.	0.118339	0.0146565	0.792275	0.92527
10.	0.118115	0.0130344	0.812122	0.943271
11.	0.117749	0.0116226	0.830208	0.95958
12.	0.117278	0.0103888	0.846687	0.974354
13.	0.116731	0.00930643	0.861702	0.987739
14.	0.11613	0.0083536	0.875381	0.999865

**Table 4 ijerph-16-00973-t004:** $\alpha_{1}=\alpha_{2}=\alpha_{3}=0.7$.

*t*	S(t)	I(t)	V(t)	Ω
0.	0.06	0.1	0.5	0.66
1.	0.0846833	0.0830972	0.513182	0.680963
2.	0.102273	0.0692909	0.52789	0.699454
3.	0.114928	0.0579974	0.543688	0.716614
4.	0.12406	0.0487393	0.560244	0.733044
5.	0.130622	0.0411297	0.5773	0.749051
6.	0.135272	0.0348562	0.594653	0.764782
7.	0.138481	0.0296671	0.612144	0.780292
8.	0.140589	0.0253599	0.629646	0.795595
9.	0.141847	0.0217719	0.647058	0.810678
10.	0.142447	0.0187717	0.664302	0.82552
11.	0.142532	0.0162536	0.681314	0.840099
12.	0.142216	0.0141322	0.698044	0.854392
13.	0.141586	0.0123381	0.714455	0.86838
14.	0.140713	0.0108151	0.730517	0.882046
15.	0.139652	0.00951743	0.746208	0.895378
16.	0.138448	0.00840763	0.761512	0.908368
17.	0.137136	0.00745506	0.776418	0.921009
18.	0.135746	0.00663451	0.790919	0.9333
19.	0.134302	0.00592521	0.805012	0.945238
20.	0.132822	0.00530997	0.818695	0.956827

**Table 5 ijerph-16-00973-t005:** $\alpha_{1}=\alpha_{2}=\alpha_{3}=0.5$.

*t*	S(t)	I(t)	V(t)	Ω
0	0.05	0.1	0.6	0.65
1.	0.0843185	0.0806307	0.603477	0.768427
2.	0.107201	0.0651439	0.610994	0.783339
3.	0.122691	0.0527997	0.621487	0.796978
4.	0.133229	0.0429688	0.634175	0.810373
5.	0.140346	0.0351331	0.648474	0.823953
6.	0.14504	0.0288752	0.663941	0.837856
7.	0.147983	0.0238632	0.680235	0.852082
8.	0.149641	0.0198347	0.697094	0.86657
9.	0.150343	0.016584	0.714313	0.88124
10.	0.150329	0.0139496	0.731731	0.896009
11.	0.149776	0.0118048	0.749222	0.910803
12.	0.148818	0.0100505	0.766688	0.925557
13.	0.147556	0.00860876	0.784051	0.940216
14.	0.146069	0.00741818	0.801249	0.954737
15.	0.144418	0.0064303	0.818236	0.969084
16.	0.142649	0.00560672	0.834975	0.98323
17.	0.140801	0.0049169	0.851436	0.997154

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
