# Peer review of "Fractional Multi-Step Differential Transformed Method for Approximating a Fractional Stochastic SIS Epidemic Model with Imperfect Vaccination"

_ijerph, 2019, doi:10.3390/ijerph16060973_

Round 1
Reviewer 1 Report
In the submitted manuscript, the authors are using a fractional multi-step differential transformed method to find approximate solutions to a fractional stochastic SIS epidemic model with imperfect vaccination. The paper is well written and correct. In my opinion it is an interesting topic and worth to publish. Nevertheless, I suggest to remove Eqs. (25), (26), (27), (28), (29), (30), (31), (32), (33) and their corresponding S(t), I(t) and V(t). There is no benefit to the reader to have these equations, it increases only the length of the paper and make it less readable. Moreover, these equations are obvious from Eqs. (16), (17), (18) and sections 5.1 and 5.2.
Author Response
The paper is well written and correct. In my opinion it is an interesting topic and worth to publish. Nevertheless, I suggest to remove Eqs. (25), (26), (27), (28), (29), (30), (31), (32), (33) and their corresponding S(t), I(t) and V(t). There is no benefit to the reader to have these equations, it increases only the length of the paper and make it less readable. Moreover, these equations are obvious from Eqs. (16), (17), (18) and sections 5.1 and 5.2.
thank you for your comments,
I delete equations (25)-(33).

Reviewer 2 Report
Hi, I think your manuscript is very good, congratulations.
I consider that presentation of solutions can be improved, I mean, as an example, page 16 solutions. Maybe in a shorter or cleaner version.
You can improve the quality and discussion of figures because one of the signal is lost ("I" bar) and your arguing is repetitive and unclear.
On the other hand, I think that you can improve the soundness of your contribution if you can explicitly compare your results with similar or equivalent recent contributions.
Best regards.
Author Response
I consider that presentation of solutions can be improved, I mean, as an example, page 16 solutions. Maybe in a shorter or cleaner version.
thank you for your comments,
In this part, I write the solutions in details, since most of the previous papers using this method didn't write any details (and that's why this method is still seldom used), as you can see I just write three steps as an example from 100 steps to give any future researcher a hint to compare his results with my results, I think a sample of 3 steps from 100 steps is a very short way.
I also, delete equations (25) to (33) to make the work shorter, since it is very clear, and other reviewers suggest to delete these equations.
You can improve the quality and discussion of figures because one of the signal is lost ("I" bar)
I improve the figures.
and your arguing is repetitive and unclear.
I add a separate section: section 6. Discussion
and add this statement, According to Section 3 and Table 2, we give the following discussions:
On the other hand, I think that you can improve the soundness of your contribution if you can explicitly compare your results with similar or equivalent recent contributions.
I really do not find any solutions to compare between my method and other methods, for two reasons, first: I think it is the first time solving this model in this large period at least from 0 to 14. Second: I think it is the first time to solve this model in fractional order.

Reviewer 3 Report
The paper needs major revisions. Please see the attached file.

Author Response
1. The authors should use the MDPI template. Now the paper is prepared using the Elsevier template (maybe earlier the paper was submitted to the Elsevier journal).
I use the MDPI template.
2. The gamma function is well known. Its definition (Equation (1)) should be deleted.
I delete this subsection.
3. On page 4, line 100, the authors state: “The Caputo fractional derivative is defined as ([5])”. I would like to emphasize that Samko et a1. [5] do not consider the Caputo derivative at all. The Caputo derivative is defined, for example, in the book of Podlubny [4].
I change this point, thank you.
4. Equation (4) contains mistakes (misprints?).
I repair these mistakes.
5. English of the paper should be improved.
I will improve it.
6. The following publications should be added to References:
Y. Povstenko, Linear Fractional Diffusion-Wave Equation for Scientists and Engineers. Birkhäuser, New York, NY, USA, 2015.
A.M.A. El-Sayed, A.A.M.Arafa, M. Khalil, A. Sayed. Backward bifurcation in a fractional order epidemiological model. Progr. Fract. Differ. Appl. 3, 281-287, 2017.
I add them.
and also, add this statement in the introduction:
The main reason for using fractional derivatives instead of integer derivatives is
that fractional order model considered the memory effect, while integer order does not give us any information about the effect of the memory of human population which influences disease transmission~\cite{38}.
7. References of the paper are prepared careless.
In Ref. [1], there are three authors: A. A. Kilbas, H.M. Srivastava, J.J. Trujillo, not only H.M. Srivastava, J.J. Trujillo. I do not understand what does it mean “215” in Ref. [1].
I modify this ref. as:
A. A. Kilbas, H.M. Srivastava, J.J. Trujillo. Theory and applications of fractional differential equations. Vol. 204, Elsevier, Amsterdam, 2006.
In Ref. [2], the place of publication (New York, NY, USA) should be added.
I add it:
K.S. Miller, B. Ross. An introduction to the fractional calculus and fractional differential equations. Wiley-Interscience, New York, NY, USA, 1993.
I do not understand what does it mean “221” in Ref. [4].
It is a mistake.
As far as I know, in Ref. [5] there should be “Gordon and Breach Science Publishers, Amsterdam, The Nethelands”, not “Switzerland”.
This book has many editions, please, open this attachment for all editions of this book.
//www.worldcat.org/title/fractional-integrals-and-derivatives-theory-and-applications/oclc/26401805/editions?referer=di&editionsView=true
In Ref. [8], there should be “Sains Malaysiana”, not "Sains Malaysina”.
I change it:
S. Abuasad, I. Hashim. Homotopy decomposition method for solving higher-order time-fractional diffusion equation via modified beta derivative. Sains Malaysiana, 47(11), (2018), 2899-2905.
In Ref. [27], which is in press, doi should be given: https://doi.org/10.1016/j.jksus.2018.02.002
ok, I add it.
Ref. [28] is not published.
I remove it.
In Refs. [30] and [31], there should be “Sci.”, not “sci.”
I repair them.
The title of Ref. [35] is wrong. There should be “Mathematical analysis of an SIS model ...”, not “Mathematical analysis of analysis model
”
I repair it.
M. Safan, F.A. Rihan. Mathematical analysis of an SIS model with imperfect vaccination and backward bifurcation. Mathematics and Computers in Simulation, 96, (2014), 195-206.

Round 2
Reviewer 3 Report
The authors have improved their paper, and I recommend it for publication.